# Plasma Neutrophil Gelatinase-Associated Lipocalin Is Useful for Predicting Mortality in Critically Ill Patients

**DOI:** 10.3390/jcm10122576

**Published:** 2021-06-10

**Authors:** Izabela Duda, Łukasz Krzych

**Affiliations:** Department of Anesthesiology and Intensive Care, School of Medicine in Katowice, Medical University of Silesia, 40-572 Katowice, Poland; l.krzych@wp.pl

**Keywords:** neutrophil gelatinase-associated lipocalin, mortality, critically ill

## Abstract

Elevated neutrophil gelatinase-associated lipocalin (NGAL) occurs in a wide range of systemic diseases. This study examined the clinical utility of plasma NGAL to predict intensive care unit (ICU) and in-hospital mortality in critically ill patients. A total of 62 patients hospitalized in a mixed ICU were included; pNGAL, creatinine, and C-reactive protein (CRP) were assayed on four consecutive days (D1-D4) following ICU admission. APACHE II score (Acute Physiology and Chronic Health Evaluation) was calculated 24 h post-admission. ICU mortality reached 35% and in-hospital mortality was 39%. The median pNGAL at admission was 142.5 (65.6–298.3) ng/mL. pNGAL was significantly higher in non-survivors compared to survivors. The highest accuracy for ICU mortality prediction was achieved at the pNGAL cutoff of 93.91 ng/mL on D4 area under the curve (AUC) = 0.89; 95%CI 0.69–0.98 and for in-hospital mortality prediction was achieved at the pNGAL cutoff of 176.64 ng/mL on D3 (AUC = 0.86; 95%CI 0.69–0.96). The APACHE II score on ICU admission predicted ICU mortality with AUC = 0.89 (95%CI 0.79–0.96) and in-hospital mortality with AUC = 0.86 (95%CI 0.75–0.94). Although pNGAL on D1 poorly correlated with APACHE II (R = 0.3; *p* = 0.01), the combination of APACHE II and pNGAL on D1 predicted ICU mortality with AUC = 0.90 (95%CI 0.79–0.96) and in-hospital mortality with AUC = 0.95 (95%CI 0.78–0.99). Maximal CRP during study observation failed to predict ICU mortality (AUC = 0.62; 95%CI 0.49–0.74), but helped to predict in-hospital mortality (AUC = 0.67; 95%CI 0.54–0.79). Plasma NGAL with combination with the indices of critical illness is a useful biomarker for predicting mortality in heterogeneous population of ICU patients.

## 1. Introduction

In the ICU, personalized specialized treatment aims to reduce mortality in critical illness, which remains high and varies between 20–40% [1,2,3]. Prognostication is an important element of a patient’s assessment post-admission and is usually based on calculations of APACHE (Acute Physiology And Chronic Health Evaluation) score, SAPS (Simplified Acute Physiology Score) score, or MPM (Mortality Probability Models) score [4].

For many years, the researchers’ interest focused on demonstrating the utility of biomarkers in a group of critically ill patients. Biomarkers are applied in various medical conditions and they change the diagnostics of many diseases, such as myocardial infarction. However, the patients hospitalized in the ICU belong to a group of heterogeneous medical conditions, therefore their application is somehow limited. Studies of biomarkers comprise two fields. The first one is used to diagnose medical conditions, the second one to predict treatment outcome. On the basis of the literature review, it can be concluded that many biomarkers applicable in prognosis have been studied, but none of them have the sufficient specificity and sensitivity to be routinely applied in clinical practice. The most thoroughly examined biomarkers in ICU patients include CRP and procalcitonin [5,6].

NGAL is a small protein weighing 25 kDa, made up of 178 amino acids. NGAL belongs to acute phase proteins and it is involved in apoptosis processes, carcinogenesis, cells differentiation, and growth. The secretion of NGAL takes place in hepatocytes, cells of renal tubules, cells of the immune system, heart, gastrointestinal tract, and respiratory system. Low NGAL expression can be determined in healthy human tissues, whereas in damaged epithelial cells it increases dramatically. Assaying NGAL concentration in serum and urine is applied in medicine. A significant NGAL increase occurs in patients with acute and chronic kidney diseases, after cardiac surgeries, in cardiovascular disease, cancer and contrast-induced nephropathy [7]. Numerous published studies evaluate diagnostic and prognostic utility of NGAL in acute kidney injury in critically ill patients [8,9,10]. NGAL expression also occurs in inflammatory conditions such as acute bacterial infections, sepsis, and septic shock. The prognostic value of NGAL in sepsis was presented in several studies and was not unambiguously verified [11,12,13,14]. Publications concerning the predictive value of NGAL in heterogeneous population of critically ill patients hospitalized in the mixed surgical-medical ICU are sparse [15,16,17,18].

We sought to conduct a prospective observation of a heterogeneous population of critically ill adults. Our specific objectives were: (1) to assess the utility of measurements of NGAL concentration in serum for predicting ICU mortality; (2) to identify NGAL concentration threshold to optimize its value for predicting the compromised outcome; (3) to establish the relation between NGAL concentration in serum and CRP, as well as APACHE II score post-ICU admission.

## 2. Materials and Methods

This prospective observational study covered critically ill adult patients hospitalized in a ten-bed mixed ICU. We screened 129 consecutive patients admitted to the ICU during a six-month period. Pregnant females (*n* = 2), patients with known chronic kidney disease, and renal replacement therapy prior to ICU admission (*n* = 8) were excluded. High mortality within the next 24 h (*n* = 4) and early ICU discharged patients (*n* = 51) was also excluded from this study. The 2016 SCCM ICU admission and discharge criteria were applied for all patients [19].

The baseline characteristics of the patients including gender, age, and primary diagnosis were recorded at admission. The severity of clinical illness was assessed by APACHE II score, based on the worst values 24 h post admission. Mortality during hospitalization in the ICU and during the hospital stay were the outcomes. Acute kidney injury (AKI) was defined using the Kidney Disease: Improving Global Outcomes (KDIGO) guidelines [20]. Sepsis was defined according to the definitions of the Society of Critical Care Medicine [21].

Blood samples were retrieved from a peripheral vein within 1 h of arrival at the ICU. Laboratory testing (pNGAL, creatinine, CRP, lactate) was performed on a daily basis for four days after admission. Samples for pNGAL containing EDTA as an anti-coagulant, were stored at −80 °C for further analysis. Plasma NGAL was measured using the BioVendor Human Lipocalin-2/NGAL ELISA (Czech Republic) following the manufacturer’s instructions. Laboratory results were reviewed collectively after study termination.

The study protocol was reviewed and approved by the Ethics Committee of the Medical University of Silesia (KNW/0022/KB/208/15; 7 October 2015). Individual informed consent was not required because the test samples were obtained from blood collected for routine laboratory tests during hospitalization in the ICU.

Statistical analysis was performed using MedCalc Statistical Software version 17.2 (MedCalc Software bvba, Ostend, Belgium). Continuous variables were expressed as median and interquartile range (IQR). Qualitative variables were expressed as absolute values and/or percent. Between-group differences for quantitative variables were assessed using Mann–Whitney U-test, after verification of variables’ distribution with Shapiro–Wilk test. Chi-squared or Fisher’s exact test were applied for qualitative variables. Correlation was assessed using the Spearman rank coefficients (R). Receiver operating characteristic (ROC) curves were drawn and the areas under the ROC curves (AUROC) were calculated to assess predictive value of investigated continuous variables. Logistic regression was applied to assess the impact of APACHE II and pNGAL on the outcome. Logistic ORs with their 95% CIs were calculated. All tests were two-tailed. ‘*p*’ value was set at 0.05.

## 3. Results

A number of 129 patients were admitted to the ICU during study period, 62 of them met the inclusion criteria. Among those individuals, 40 survived (65%) and 22 died (35%) during the ICU stay. Eight patients died on day two, four patients on day three, and five patients on day four of the ICU stay. In-hospital mortality reached 39% (*n* = 24). The baseline characteristics and clinical features, with differences between survivors and non-survivors, are shown in Table 1.

The median concentration of pNGAL at admission was 142.5 (65.6–298.3) ng/mL (Table 2). The median CRP at ICU admission was 70.5 (12.84–165.96) mg/L. pNGAL, creatinine, CRP, and lactate concentrations at admission were statistically significantly higher in non-survivors.

Moreover, pNGAL was significantly higher in non-survivors compared to survivors on each of the four days of observation, and the highest difference was observed on day one (Table 3).

On each of the four study days, pNGAL was confirmed to be a significant predictor of mortality; however, its concentration on day four was the most powerful, i.e., AUC = 0.89; 95%CI 0.69–0.98, with sensitivity = 100%, specificity = 77%, Youden index = 0.77, positive likelihood ratio = 4.33, negative likelihood ratio = 0, using a cutoff of 93.91 ng/mL (Figure 1). For in-hospital mortality, the best prediction was achieved at the pNGAL cutoff of 176.64 ng/mL on D3 (AUC = 0.86; 95%CI 0.69–0.96) (with sensitivity = 84%, specificity = 78%, Youden index = 0.62, positive likelihood ratio = 3.81, negative likelihood ratio = 0.2) (Figure 2).

APACHE II score was also a significant mortality predictor (ICU mortality: AUC = 0.89; 95%CI 0.79–0.96, with sensitivity = 86% and specificity = 87% using a cutoff of 20 points/in-hospital mortality: AUC = 0.86; 95%CI 0.75–0.94, with sensitivity = 75% and specificity = 84% using a cutoff of 20 points) (Figure 3A,B).

pNGAL at admission poorly correlated with APACHE II score (R = 0.31; *p* = 0.01). The correlation was stronger at day five (R = 0.52; *p* < 0.01). Strong positive correlation was reported between pNGAL and creatinine from day one till day four. The correlation between pNGAL and CRP was strong only on day one and day two. No correlation was found between pNGAL and lactate, except for day two (Table 4).

pNGAL differed between patients who required renal replacement therapy and those without this treatment on day one, three, and four, and CRP differed between subjects only post-ICU admission (Table 5). All therapies were started at day one.

Combination of APACHE II (OR = 1.41; 95%CI 1.14–1.74) and pNGAL (OR = 1.01; 95%CI 1–1.01) on day one were acceptable ICU mortality predictors. The combination of APACHE II and pNGAL on day one predicted ICU mortality with AUC = 0.90 (95%CI 0.79–0.96) and in-hospital mortality with AUC = 0.95 (95%CI 0.78–0.99).

CRP concentrations from day one till day four were impractical in ICU prognostication (*p* > 0.05 for all). Maximal CRP during study observation also failed to predict ICU mortality (AUC = 0.62; 95%CI 0.49–0.74), but helped to predict in-hospital mortality (AUC = 0.67; 95%CI 0.54–0.79). Also, CRP concentrations on day one and two were statistically significant predictors of in-hospital mortality: AUC = 0.81 (95%CI 0.69–0.90), *p* < 0.001 and AUC = 0.70 (95%CI 0.56–0.81), *p* = 0.01.

## 4. Discussion

This study evaluated utility of pNGAL levels in predicting ICU mortality. We showed NGAL level on each of the four consecutive days of ICU hospitalization from the time of admission. On each day, we assessed the predictive value of NGAL in predicting ICU mortality. On each consecutive day, a statistically significant correlation between NGAL level and ICU mortality was reported, based on AUC. The most distinct correlation was observed on the fourth day of hospitalization. Moreover, the correlation between NGAL level and ICU mortality at admission was also significantly strong. This observation suggests the prognostic value of assaying NGAL level at ICU admission in critically ill patients. The application of NGAL for predicting ICU mortality does not have the extensive literature and the published observations are also ambiguous. In some data the correlation between NGAL and mortality is emphasized, whereas other data lack this correlation. Haase analyzed seven NGAL trials for predicting mortality [22]. Based on ROC curve, he concluded that NGAL can be a useful prognostic tool, but with certain limitations, with respect to in-hospital mortality (AUC = 0.706). Hjortrup analyzed six NGAL trials for predicting mortality based on ROC curve [23]. Three trials concerned assaying plasma NGAL and the other three urinary NGAL. Plasma NGAL appeared to be a poor mortality predictor (AUC range 0.58–0.67). In contrast to those studies, Mahmoodpoor et al. found the important value of plasma NGAL 48 h after admission for predicting ICU mortality (AUC = 0.874) [24]. Shavit et al. examined plasma NGAL level before and during non-cardiac major surgery [15]. They found that NGAL is a predictor of infection and in-hospital death. Hang et al. found a similar correlation in the study conducted in critically ill patients in the emergency department. ROC curve for predicting 28-day mortality was 0.723 [25].

NGAL cutoff values for mortality, presented in publications, cover a broad range from 80 to 480 ng/mL [22]. In our study, cutoff was the lowest on the fourth day of trial and amounted to 93.91 ng/mL. It is a relatively low value compared to other studies, for example Kyner’s, where cutoff was indicated as >480 ng/mL [26]. However, Shavit published a similarly low cutoff value (98 ng/mL) [15]. The differences may result from various assaying techniques (ELISA vs. fluorescence immunoassays), storing temperature, and sample type (serum or plasma). Assaying NGAL in serum is preferred rather than in plasma because during centrifugation releases of NGAL from neutrophils can occur.

Our study demonstrates a strong correlation between CRP level and NGAL level on the first two days. However, in the next two days the study shows the lack of any correlation. Moreover, we did not determine a predictive CRP value for predicting ICU mortality. CRP secretion occurs 4–6 h after stimulation, and the top release occurs at 39 h from the beginning of inflammatory reaction. Therefore, for the study we selected the maximum level of CRP. Likewise, it was found in several studies where CRP was assessed as a mortality predictor according to AUC, which was low [27,28,29]. Other studies, however, showed a significant correlation between CRP concentration in plasma and mortality predictor in the ICU [30,31]. Those studies concerned patients diagnosed with sepsis. The presented study concerns the group of critically ill patients, including 38% treated for sepsis. Park and al. evaluated the predictive value of CRP at admission in critically ill patients in the medical ICU. There were 70% of patients with sepsis. AUC of CRP for mortality in ICU patients was 0.576 [32]. In another study conducted in the mixed medical-surgical ICU, CRP level measured at discharge was not a predictor of readmission or death [33]. The application of CRP for prognosis in critically ill patients still remains controversial.

APACHE II score is a widely accepted tool for calculating the risk of death. The correlation between scoring and the risk of death was indicated in many studies [4,34,35,36]. In our study, APACHE II was a strong predictor of ICU mortality (AUC = 0.893). At the same time, we demonstrated a positive correlation between NGAL level at admission to the ICU and APACHE II score. This is a further indication of a substantial value of NGAL in predicting outcomes.

However, the study has certain limitations. Firstly, there are differences in baseline severity and therapeutic activities among ICU patients, which precludes forming a homogenous study group in terms of diagnostics, monitoring, and therapy. Secondly, the study was conducted in one ICU ward, therefore it can be potentially susceptible to error and limited statistical capability. Thirdly, due to the strict inclusion criteria, it cannot be ruled out that a greater number of patients may weaken the correlation between NGAL and ICU mortality.

There are several strengths to our study. Firstly, our study is one of the few studies that evaluate a heterogeneous group of critically ill patients admitted to the ICU for medical and surgical reasons. Secondly, the study was not restricted to a single assay at admission, but it was conducted on four consecutive days of hospitalization, which enhances the credibility of the presented results. Thirdly, there is a variety of other prognostic factors of ICU mortality, many of which are assayed in APACHE score. Therefore, in order to strengthen the relation of NGAL and ICU mortality, the result was correlated with mortality assessment in APACHE score. Finally, RRT use might influence our observations regarding pNGAL and CRP concentrations. Both molecules are easily eliminated and it may introduce a bias and invalidate the results. We sought to minimize this issue by performing additional sub-analyses.

## 5. Conclusions

Plasma NGAL, in combination with the indices of critical illness (i.e., baseline APACHE II score) is a useful biomarker for predicting mortality in a heterogeneous population of ICU patients. pNGAL is more powerful in prognostication than CRP. The role of NGAL in critical illness should be verified in further research.

## Figures and Tables

**Figure 1 jcm-10-02576-f001:**
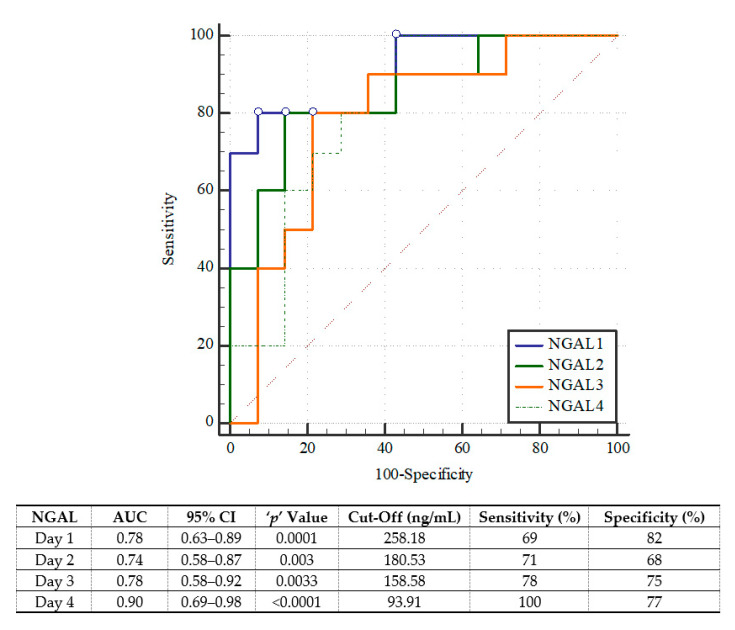
The receiver operating characteristic curves of plasma neutrophil gelatinase-associated lipocalin (NGAL) in prediction of ICU mortality.

**Figure 2 jcm-10-02576-f002:**
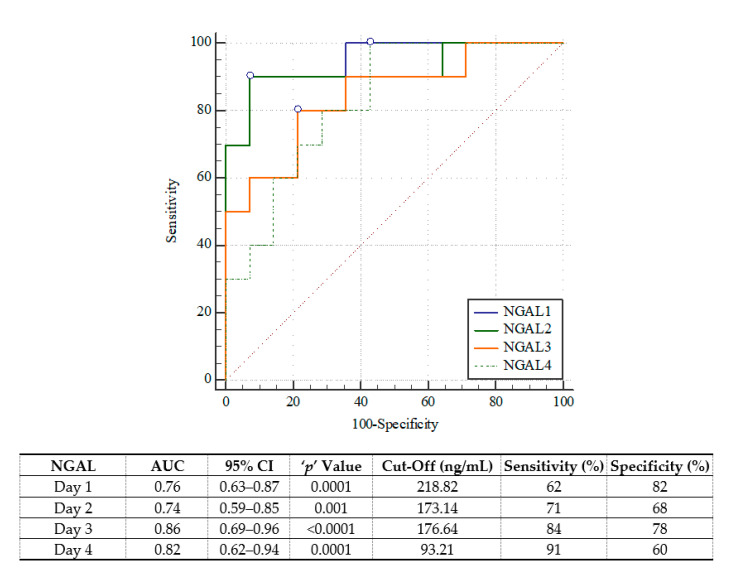
The receiver operating characteristic curves of plasma neutrophil gelatinase-associated lipocalin (NGAL) in prediction of in-hospital mortality.

**Figure 3 jcm-10-02576-f003:**
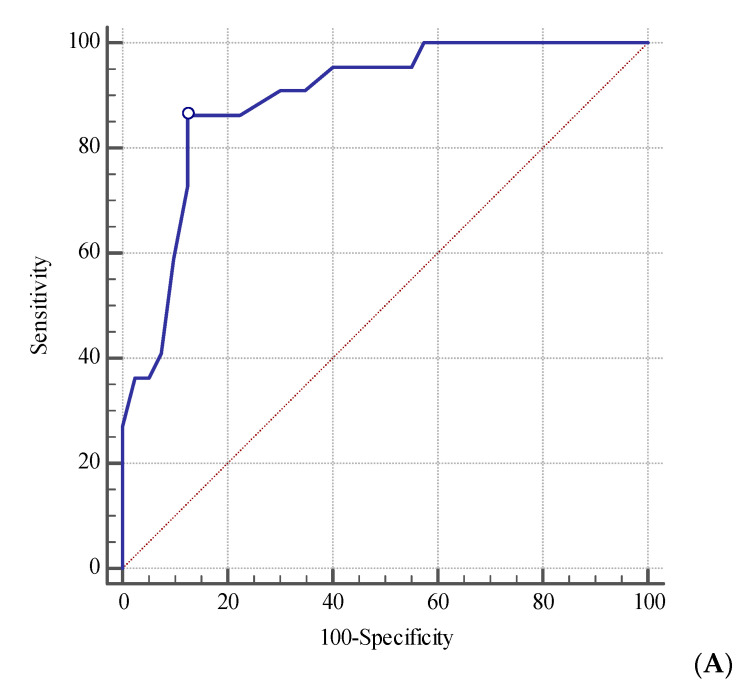
The receiver operating characteristic curve for baseline APACHE II score in prediction of intensive care unit (**A**) and in-hospital (**B**) mortality.

**Table 1 jcm-10-02576-t001:** Baseline characteristics and clinical features of the enrolled patients.

Variable	Value	ICU		In-Hospital	
		Survivors	Non-Survivors	Survivors	Non-Survivors
Age, years (median, IQR)	59 (43–67)	50 (39–63)	72 (61–82) *	50 (37–64)	66 (60–81) *
Male gender, *n* (%)	27 (43)	9 (33)	18 (67)	10 (37)	17 (63)
APACHE II score (median, IQR)	18 (12–23)	14 (9–18)	23 (21–28) *	14 (9–18)	23 (20–28) *
ICU length of stay, day (median, IQR)	9 (4–21)	9 (4–16)	10 (4–29)	-	-
Hospitalization category, *n* (%)					
Medical	32 (51)	25 (78)	7(21) *	24 (75)	8 (25) *
Surgical	30 (49)	15 (50)	15 (50)	14 (46)	16 (53)
Sepsis, *n* (%)	24 (38)	11 (35)	13 (65) *	10 (42)	14 (58) *
Acute kidney injury, *n* (%)	15 (24)	7 (47)	8 (53)	5 (33)	10 (67) *
Vasopressor use, *n* (%)	39 (63)	19 (49)	20 (51) *	17 (44)	22 (56) *
Mechanical ventilation, *n* (%)	55 (89)	33 (60)	22 (40)	31 (56)	24 (44)
Renal replacement therapy, *n* (%)	7 (11)	2 (28)	5 (72) *	1 (14)	6 (86) *

APACHE: acute physiology and chronic health evaluation; ICU: intensive care unit. * *p* < 0.05.

**Table 2 jcm-10-02576-t002:** Laboratory analyses of patients on ICU admission.

Variable	Values	ICU		In-Hospital	
		Survivors	Non-Survivors	Survivors	Non-Survivors
NGAL (ng/mL)	142.5 (65.62–298.29)	88.8 (45–183)	297.0 (102–339) *	88.8 (48–150)	279.4 (116–326) *
Creatinine (mg/dL)	1.08 (0.82–1.61)	0.89 (0.77–1.22)	1.52 (1.21–2.91) *	0.89 (0.78–1.18)	1.83 (1.1–3.42) *
CRP (mg/L)	70.5 (12.84–165.96)	27.8 (5.63–149)	146.6 (65.5–274.8) *	18.54 (5.57–145)	146.6 (73.6–278.1) *
Lactate (mmol/L)	2.1 (1.20–3.10)	1.75 (1–2.85)	2.45 (1.5–3.4) *	1.6 (1–2.8)	2.45 (1.6–3.5) *

NGAL: neutrophil gelatinase-associated lipocalin; CRP: C-reactive protein. Vales are medians (IQR). * *p* < 0.05.

**Table 3 jcm-10-02576-t003:** Neutrophil gelatinase-associated lipocalin (NGAL) in survivors and non-survivors.

	ICU		In-Hospital	
NGAL (ng/mL)	Survivors (*n* = 40)	Non-Survivors (*n* = 22)	Survivors	Non-Survivors
Day 1	88 (45–183)	297 (102–339) *	88.8 (48–150)	279.4 (116–326) *
Day 2	131 (61–274)	251 (151–360) *	131.4 (63.5–188.4)	283.7 (149–360) *
Day 3	111 (68–248)	271 (213–354) *	102.6 (68.4–173)	297 (224–373.6) *
Day 4	75 (66–131)	250 (142–349) *	91.2 (64.6–162)	239 (130–348) *

Data expressed as median (interquartile range). * *p* < 0.05.

**Table 4 jcm-10-02576-t004:** Correlation between neutrophil gelatinase-associated lipocalin (NGAL) levels and baseline C-reactive protein (CRP), creatinine, lactate, and APACHE II.

	Day 1	Day 2	Day 3	Day 4
CRP (mg/L)	R = 0.774 *p* < 0.0001	R = 0.589 *p* < 0.0001	R = 0.269 *p* = 0.1755	R = 0.214 *p* = 0.2946
Creatinine (mg/dL)	R = 0.517 *p* < 0.0001	R = 0.543 *p* = 0.0001	R = 0.733 *p* < 0.0001	R = 0.857 *p* < 0.0001
Lactate (mmol/L)	R = 0.226 *p* = 0.1038	R = 0.430 *p* = 0.0032	R = 0.396 *p* = 0.0335	R = 0.367 *p* = 0.0649
APACHE II (points)	R = 0.309 *p* = 0.0195	R = 0.256 *p* = 0.0726	R = 0.322 *p* = 0.0772	R = 0.518 *p* = 0.0057

Data expressed as Spearman coefficient of correlation and ‘*p*’ value.

**Table 5 jcm-10-02576-t005:** Neutrophil gelatinase-associated lipocalin (NGAL) C-reactive protein (CRP) levels in terms of renal replacement therapy use.

	Renal Replacement Therapy (−)	Renal Replacement Therapy (+)
NGAL day 1 (ng/mL)	102.7 (64.3–281.7)	279.4 (235.8–318.6) *
NGAL day 2 (ng/mL)	151.5 (72.2–282.6)	316.5 (186.5–327.4)
NGAL day 3 (ng/mL)	145.8 (76.4–256.1)	310.2 (229.4–368.8) *
NGAL day 4 (ng/mL)	116.3 (66.4–227.8)	250.4 (200.6–357.7) *
CRP day 1 (mg/L)	45.2 (9.4–158)	238 (61.3–348) *
CRP day 2 (mg/L)	75.6 (34.4–168.7)	268 (76.2–357.5)
CRP day 3 (mg/L)	92.5 (42.6–178.1)	192 (77–290.2)
CRP day 4 (mg/L)	104.3 (57–170.6)	145.6 (64.3–189)

Vales are medians (IQR). * *p* < 0.05.

## Data Availability

The data presented in this study are available on request from the corresponding author.

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
