# Peer review of "Plasma Neutrophil Gelatinase-Associated Lipocalin Is Useful for Predicting Mortality in Critically Ill Patients"

_jcm, 2021, doi:10.3390/jcm10122576_

Round 1
Reviewer 1 Report
I think it's a creative study. However, there are several questions about the method and result. Study populations with heterogeneous characteristics have many biases, so a large volume of the study population is required, and several variables must be considered in the analysis.
Introduction
Line 59. I am wondering why you analyzed ICU mortality, not in-hospital mortality.
Line 60. The authors referred to CRP and procalcitonin as the most thoroughly examined biomarkers. Among them, I am curious about the reason for choosing CRP.
Materials and method
Line 68. What are the exact criteria for early ICU discharge patients in the exclusion criteria?
Are pediatric patients included in this study?
Line 77. Is there a reason for not including procalcitonin in laboratory testing?
Line 78. In this study, patients who died within 24 hours were excluded from the study. So, how did you deal with the data of patients who died within 2 to 4 days after hospitalization?
Result
Line 98. Total enrolled patients are '62', not '64'.
Table 1. Even considering the study conducted on heterogenous patients, it is important to understand the characteristics of the patient group. A specific diagnosis or treatment method (mechanical ventilation, central venous catheterization, CRRT, operation, radiologic intervention, etc.) will need to be added. In addition, comparison between survivor vs non-survivor of the variable mentioned in Table 1 is considered necessary.
Table 2. Are the values presented in the table above the result on the day of admission to the ICU?
Table 3. I wonder why pNGAL was observed up to day 4 unlike other laboratory results.
Figure 1. Day 4 NGAL had the highest AUC of 0.90, but was there any analysis on the values on or after day 5? In addition, it would be better to present PPV and NPV in addition to sensitivity and specificity.
Line 125. The correlation between pNGAL and APACHE II scores can be presented in Table 4 to better understand the results.
Line 132. Why did you analyze the combination of APACHE II and pNGAL on day 1?
Discussion
Line 142~163. The authors tried to predict ICU mortality, not in-hospital mortality. However, the references are studies of in-hospital mortality. It is necessary to present the clinical significance of predicting ICU mortality.
Line 165~170. Why did you mention Urine NGAL?
Line 171~178. There are no data on the timing of death of patients with ICU deaths in this study. Excluding this, there is a bias to talk about the AUC and cut-off values for the ICU mortality of day 4 NGAL.
Line 198~199. The results section (line 125) suggested that pNGAL at admission and APACHE II score were poorly correlated. This is an interpretation contrary to the previous result.
Conclusion
Line 217~218. It was said that baseline critical status (i.e. APACHE II score) was adjusted, but it was not mentioned in the method and results. How did you adjust it?
Author Response
Rev#1
Thank you for your valuable comments and suggestions.
Q1. I am wondering why you analyzed ICU mortality, not in-hospital mortality.
R1. We corrected the paper. Additional analysis regarding in-hospital mortality was performed.
Q2. The authors referred to CRP and procalcitonin as the most thoroughly examined biomarkers. Among them, I am curious about the reason for choosing CRP.
R2. CRP is regarded as one of the most important acute phase proteins.
Sproston NR, Ashworth JJ. Role of C-Reactive Protein at Sites of Inflammation and Infection. Front Immunol. 2018 Apr 13;9:754. doi: 10.3389/fimmu.2018.00754. PMID: 29706967; PMCID: PMC5908901
Q3. What are the exact criteria for early ICU discharge patients in the exclusion criteria?
R3. 2016 SCCM criteria were applied for all admissions and discharges (lines 76-77).
Q4. Are pediatric patients included in this study?
R4. Only adult patients were included (line 71).
Q5. Is there a reason for not including procalcitonin in laboratory testing?
R5. Nowadays Procalcitonin is used as a tool to guide antibiotic discontinuation
Bouadma L, Luyt CE, Tubach F, Cracco C, Alvarez A, Schwebel C, Schortgen F, Lasocki S, Veber B, Dehoux M, Bernard M, Pasquet B, Régnier B, Brun-Buisson C, Chastre J, Wolff M; PRORATA trial group. Use of procalcitonin to reduce patients' exposure to antibiotics in intensive care units (PRORATA trial): a multicentre randomised controlled trial. Lancet. 2010 Feb 6;375(9713):463-74. doi: 10.1016/S0140-6736(09)61879-1. Epub 2010 Jan 25. PMID: 20097417.
Q6. In this study, patients who died within 24 hours were excluded from the study. So, how did you deal with the data of patients who died within 2 to 4 days after hospitalization?
R6. These data were analyzed according to study protocol. Deceased patients on day 2 were lost-to-follow-up observation for day 3/4 and patients who died on day 3 were lost-to-follow-up observation for day 4.
Q7. Total enrolled patients are '62', not '64'.
R7. Corrected.
Q8. Table 1. Even considering the study conducted on heterogenous patients, it is important to understand the characteristics of the patient group. A specific diagnosis or treatment method (mechanical ventilation, central venous catheterization, CRRT, operation, radiologic intervention, etc.) will need to be added. In addition, comparison between survivor vs non-survivor of the variable mentioned in Table 1 is considered necessary.
R8. We included data regarding ventilation and RRT. No operations were performed. No radiologic interventions were performed. Comparison was done.
Q9. Table 2. Are the values presented in the table above the result on the day of admission to the ICU?
R9. Yes.
Q10. Table 3. I wonder why pNGAL was observed up to day 4 unlike other laboratory results.
R10. It was performed in accordance with our study protocol. To be honest, we had financial reimbursement for only 4 measurements.
Q11. Figure 1. Day 4 NGAL had the highest AUC of 0.90, but was there any analysis on the values on or after day 5? In addition, it would be better to present PPV and NPV in addition to sensitivity and specificity.
R11. We calculated Youden index, positive and negative likelihood ratios for selected cutoff points. PPV and NPV cannot be calculated for ROC curves. They are typical parameters used to express 2x2 tables for qualitative data. There was no analysis post-day 4.
Q12. The correlation between pNGAL and APACHE II scores can be presented in Table 4 to better understand the results.
R12. Done.
Q13. Why did you analyze the combination of APACHE II and pNGAL on day 1?
R13. APACHEII is an acknowledged method of assessment of critical illness on ICU admission. Therefore we sought to verify if combination of both markers (i.e. biomarker = pNGAL, and mathematical score = APACHE II) can modify prognostication. We found it could be helpful to integrate all data.
Q14. The authors tried to predict ICU mortality, not in-hospital mortality. However, the references are studies of in-hospital mortality. It is necessary to present the clinical significance of predicting ICU mortality.
R14. We corrected analysis and included in-hospital mortality.
Q15. Why did you mention Urine NGAL?
R15. We deleted it.
Q16. There are no data on the timing of death of patients with ICU deaths in this study. Excluding this, there is a bias to talk about the AUC and cut-off values for the ICU mortality of day 4 NGAL.
R17. We included data regarding the day of death (line 110-111).
Q18. The results section (line 125) suggested that pNGAL at admission and APACHE II
score were poorly correlated. This is an interpretation contrary to the previous result.
R18. Yes, you are right. The correlation was poor, according to the statistical interpretation of the Spearman’s rank coefficient. But the diagnostic accuracy, according to the statistical interpretation of the ROC curve analysis, was good.
Q19. It was said that baseline critical status (i.e. APACHE II score) was adjusted, but it was not mentioned in the method and results. How did you adjust it?
R19. Conclusions were corrected.
Reviewer 2 Report
This is an interesting study tackling the issue of plasma neutrophil gelatinase-associated lipocalin is useful 3 for predicting mortality in critically ill patients…”
Major Comments :
1)As you have almost 25 % with AKI , half of these will be on RRT.Indeed, patients with known chronic kidney diseaese and renal replacement therapy prior to ICU admission (n=8) were excluded…Prior to ICU admission but not during…as you said NGAL is a small protein weighing 25kDa……The cut-off of the membrane used during RRT is about 35,0000 up 40,0000 Da and so NGAL will be easily eliminated and a falsely low NGAL due to RRT elimination…will introduce a bias and may invalidate the results…
2)Nowadays, SAPS III is much better severity score in Enrope as compared to APACHE II which is nowadays almost useless to evaluate the risk of death…
3)Your study is severely underpowered to evaluate mortality in relationship with NGAL either in the plasma or in the urine…
5-CRP is a pentamer of 125,000 dalton made of 5 monomers and it is the CRP monomeric which is present in the blood of septic patients .With a molecular off 22,000 Da, CRP monomeric will be easily eliminated by RRT.
The cut-off of the membrane used during RRT is about 35,0000 up 40,0000 Da and so CRP monomeric will be easily eliminated and a falsely low mCRP due to RRT elimination…will introduce a bias and may invalidate again the results…
Author Response
Rev#2.
Thank you for your valuable comments and suggestions.
Q1. As you have almost 25 % with AKI , half of these will be on RRT.Indeed, patients with known chronic kidney diseaese and renal replacement therapy prior to ICU admission (n=8) were excluded…Prior to ICU admission but not during…as you said NGAL is a small protein weighing 25kDa……The cut-off of the membrane used during RRT is about 35,0000 up 40,0000 Da and so NGAL will be easily eliminated and a falsely low NGAL due to RRT elimination…will introduce a bias and may invalidate the results…
R1. We included data regarding RRT use (table 1). We verified pNGAL concentrations between patients with and without RRT (table 5). We include this issue in limitations.
Q2. Nowadays, SAPS III is much better severity score in Enrope as compared to APACHE II which is nowadays almost useless to evaluate the risk of death…
R2. We calculated APACHEII because this scoring (not SAPS III) is recommended by the national consultant in anesthesiology and intensive care medicine.
Q3. Your study is severely underpowered to evaluate mortality in relationship with NGAL either in the plasma or in the urine…
R3. Our study was limited by enrolling small number of patients. Other authors that we cited in our manuscript tested similar small group of patients, e.g. Shavit et al. – 74 patients [15],Mahmoodpoor et al. – 50 patients [24], Martensson et al. – 45 patients [23].
Q4. 5-CRP is a pentamer of 125,000 dalton made of 5 monomers and it is the CRP monomeric which is present in the blood of septic patients .With a molecular off 22,000 Da, CRP monomeric will be easily eliminated by RRT. The cut-off of the membrane used during RRT is about 35,0000 up 40,0000 Da and so CRP monomeric will be easily eliminated and a falsely low mCRP due to RRT elimination…will introduce a bias and may invalidate again the results…
R4. We included data regarding RRT use (table 1). We verified CRP concentrations between patients with and without RRT (table 5). We include this issue in limitations.
This manuscript is a resubmission of an earlier submission. The following is a list of the peer review reports and author responses from that submission.